# Dual catalysis for enantioselective convergent synthesis of enantiopure vicinal amino alcohols

Chen-Xi Ye [1], Yared Yohannes Melcamu[1,2,3], Heng-Hui Li[1], Jiang-Tao Cheng[1], Tian-Tian Zhang[1], Yuan-Ping Ruan[1], Xiao Zheng[1,4], Xin Lu[1,2,3] & Pei-Qiang Huang [1,2]

Enantiopure vicinal amino alcohols and derivatives are essential structural motifs in natural products and pharmaceutically active molecules, and serve as main chiral sources in asymmetric synthesis. Currently known asymmetric catalytic protocols for this class of compounds are still rare and often suffer from limited scope of substrates, relatively low regio- or stereoselectivities, thus prompting the development of more effective methodologies. Herein we report a dual catalytic strategy for the convergent enantioselective synthesis of vicinal amino alcohols. The method features a radical-type Zimmerman–Traxler transition state formed from a rare earth metal with a nitrone and an aromatic ketyl radical in the presence of chiral N,N'-dioxide ligands. In addition to high level of enantio- and diastereoselectivities, our synthetic protocol affords advantages of simple operation, mild conditions, high-yielding, and a broad scope of substrates. Furthermore, this protocol has been successfully applied to the concise synthesis of pharmaceutically valuable compounds (e.g., ephedrine and selegiline).

[1] Department of Chemistry and Fujian Provincial Key Laboratory of Chemical Biology, College of Chemistry and Chemical Engineering, Xiamen University, Xiamen, Fujian 361005, China. [2] Collaborative Innovation Center of Chemistry for Energy Materials, Xiamen University, Xiamen, Fujian 361005, China. [3] State Key Laboratory of Physical Chemistry of Solid Surfaces, Xiamen University, Xiamen, Fujian 3 361005, China. [4] Key Laboratory of Synthetic Chemistry of Natural Substances, Shanghai Institute of Organic Chemistry, Chinese Academy of Sciences, Shanghai 200032, China. Correspondence and requests for materials should be addressed to X.Z. (email: zxiao@xmu.edu.cn) or to X.L. (email: xinlu@xmu.edu.cn) or to P.-Q.H. (email: pqhuang@xmu.edu.cn)

**E**nantiopure vicinal amino alcohols and their derivatives represent one of the most significant synthetic building blocks and key subunits of pharmaceutically active molecules, chiral auxiliaries and ligands. Synthesis of such compounds has stimulated continuing interest and extensive efforts[1–9]. Traditional methods for this aim are such addition reactions that mostly require enantiopure substrates or reagents, including functional group transformation of vicinal *N,O*-compounds[1–3], addition of *N/O*-heteroatoms to substrates[4–7], and nitro-group's derivatization via nucleophilic nitroaldol (Henry) reaction[8] (**I–III** in Fig. 1a). However, these strategies, including their catalytic enantioselective evolutions[3,6–8], suffered from either structurally limited substrates/products or relatively low regioselectivity.

Compared with the above protocols, radical cross-coupling between amine and alcohol moieties represents an inherently efficient and flexible way for construction of vicinal amino alcohols (Fig. 1a-**IV**). By using SmI$_2$ as reductant and oxophilic coordination center, reductive cross-coupling of imine derivatives[9–11] or nitrones[12–15] with carbonyl compounds (e.g., aldehydes/ketones) allows for an easy access to these compounds with various structures. However, the use of SmI$_2$ in stoichiometric quantity poses a

substantial challenge for enantioselective induction from chiral ligands[16], along with unavoidable side reactions such as pinacol-type homocoupling and reduction of substrates (Fig. 1b). Recently, photocatalysis[17–22] also provided several schemes on vicinal amino alcohols and their derivatives[23–28], including three enantioselective protocols catalyzed by photocatalyst-merged dual catalyst systems with chiral phosphoric acid organocatalyst[23] or chiral rhodium Lewis acid[24], as well as bifunctional Lewis acid/photoredox catalyst[25] of chiral-at-metal iridium complex[29]. Nevertheless, all of these methods relied heavily on specially designed substrates.

We recently envisioned that an efficient and flexible strategy for enantiopure vicinal amino alcohols might be realized by aptly combining merits of the aforementioned SmI$_2$-mediated[12–15] and photocatalytic[23–28] protocols from nitrones and aldehydes. That is, a photocatalytic protocol featuring with intermolecular single-electron-transfer (SET) can be used to reduce selectively the substrate of higher electron affinity (i.e., higher reduction potential)[26,30] and an oxophilic Lewis acid co-catalyst (e.g., rare earth metal cation) can be introduced to bind simultaneously both substrates[31,32], and, more importantly, an appropriate chiral ligand to form precursor complex **I** and to induce desired enantioselectivity likely through a radical-type

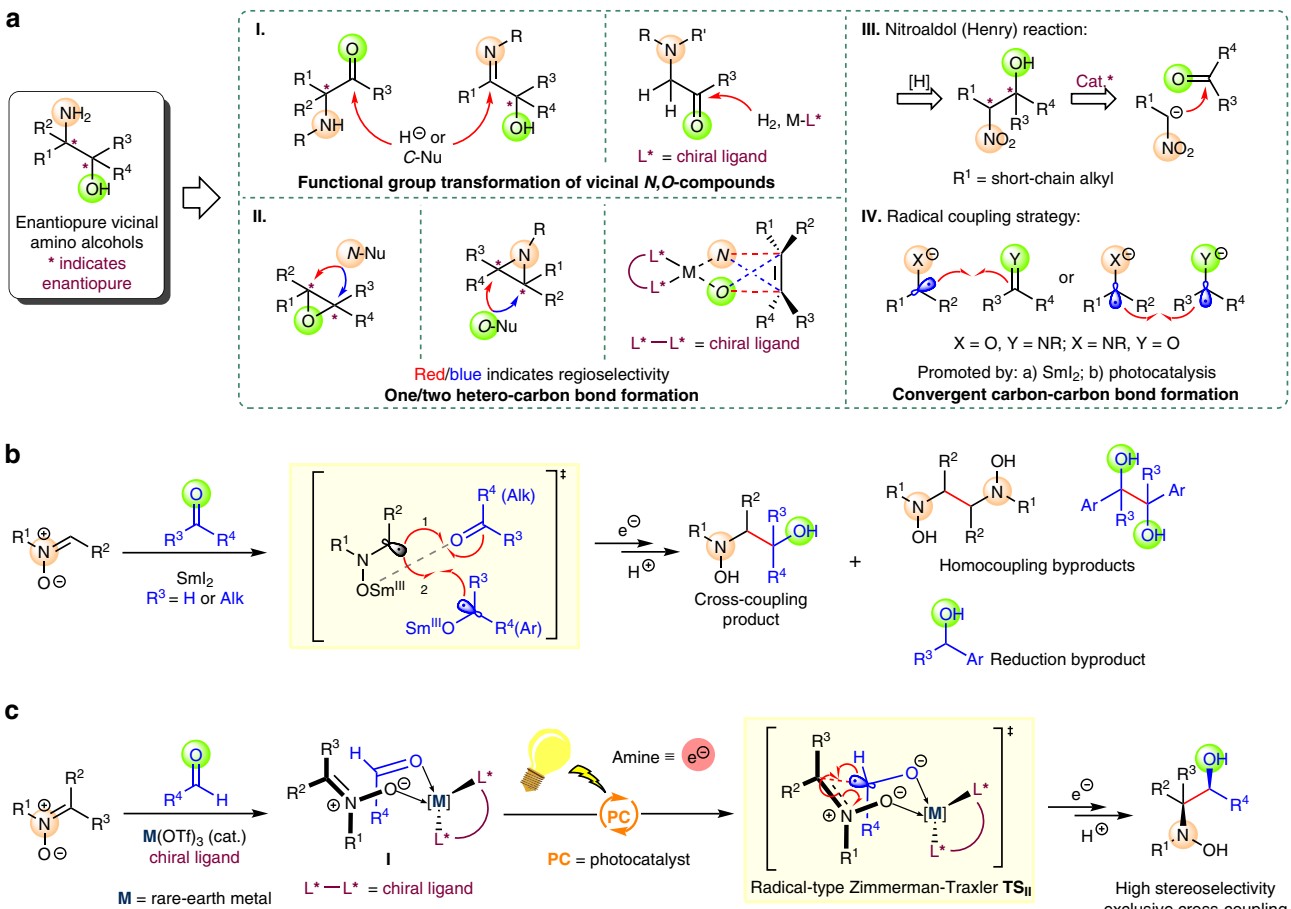

**Fig. 1** Retrosynthetic analysis and synthetic protocols of enantiopure vicinal amino alcohols. **a** The general protocols for synthesis of enantiopure vicinal amino alcohols and their derivatives. **b** Previous studies by Py and our group on SmI$_2$-mediated cross-coupling of nitrones with aldehydes/ketones may lead to homocoupling and reduction byproducts and are not ideally suitable for developing catalytic enantioselective variant. **c** Enantioselective reductive cross-coupling reaction of nitrones with aromatic aldehydes via the synergistic catalysis of chiral ligand-coordinated Lewis acid and photocatalyst was described. Through the radical-type Zimmerman–Traxler **TS$_{II}$**, vicinal hydroxyamino alcohols could be obtained exclusively with high stereoselectivity. This mild reaction is operationally simple with a wide array of nitrones and aromatic aldehydes

**Table 1 Photocatalytic enantioselective synthesis of vicinal hydroxyamino alcohol 2a-1**

| Entry | Lewis acid | Chiral ligand (mmol%) | Co-reductant | Solvent | Yield[a] (%) | dr[b] | ee[c] (%) |
|---|---|---|---|---|---|---|---|
| 1 | Sc(OTf)$_3$ | **L1-a (18)** | TEEDA | DCM | 81 | 1/10 | 89 |
| 2 | Sc(OTf)$_3$ | **L1-b** (18) | TEEDA | DCM | 73 | 1/11 | 92 |
| 3 | Sc(OTf)$_3$ | **L1-b** (18) | TEEDA | DCE | 93 | 1/12 | 92 |
| 4 | La(OTf)$_3$ | **L2-a** (30) | DIPEA | CH$_3$CN | 73 | 1/5.7 | − 16 |
| 5 | La(OTf)$_3$ | **L2-b** (30) | DIPEA | CH$_3$CN | 83 | 1/6.9 | − 24 |

[a] The reactions were performed on 0.3 mmol scale of nitrone **1a**, yields were determined by [1]H NMR analysis using 1,3,5-trimethoxybenzene as an internal standard
[b] dr values were detected from [1]H NMR analysis of crude products ($\delta_H$ 5.53, 4.87 in CDCl$_3$)
[c] ee values were detected from chiral HPLC analysis of the major diastereo isomer
* bpy, 2,2′-bipyridyl; CFL, compact fluorescent lamp; DCE, 1,2-dichloroethane; DCM, dichloromethane; i-Bu, isobutyl; i-Pr, isopropyl;
* TEEDA, N,N,N′,N′-tetraethylethylenediamine; DIPEA, N,N-diisopropylethylamine

Zimmerman–Traxler transition state **TS$_{II}$**[33,34] in the subsequent cross-coupling. Herein we report a synergistic catalysis of chiral N,N′-dioxides (Feng's ligands)[35,36] coordinated rare earth ion and Ru-photocatalyst for enantioselective radical convergent synthesis of enantiopure vicinal hydroxyamino alcohols from nitrones and aromatic aldehydes, together with a mechanism deciphering catalytic cycle and stereoselectivity of this reaction.

## Results

**Optimization of the reaction conditions.** Our investigation into this dual catalysis protocol began by studying the model reaction of nitrone **1a** with 4-fluorobenzaldehyde under various conditions (Table 1). Initial experiments revealed the synergistic catalytic effect of Lewis acid and photocatalyst, which led to desired vicinal hydroxyamino alcohol (±)− **2a-1**. Among several chiral ligand classes examined (see Supplementary Table 1), Feng's chiral N,N′-dioxides (such as **L1-a** and **L1-b**), which are well-known privileged chiral catalysts[37], provided enantioselective results for this reductive coupling reaction (see Supplementary Tables 2 and 3), despite that this class of ligands has not proved to be effective for asymmetric radical reaction before[35,36]. Meanwhile, we noted that chiral N,N′-dioxides were potentially reduced by this dual-catalyst system owing to the oxidation ability of tertiary amine oxides (see Supplementary Note 1 for details). Fortunately, by optimization of Lewis acids we found that rare earth ions coordinated N,N′-dioxides were stable enough under our conditions to give optically enriched product. PyBOX ligands, which are originally utilized by Yoon and colleagues[38] for photocatalyst/chiral Lewis acid dual-activation catalysis, also provided enantioselectivity but only with low ee (such as **L2-a** and **L2-b**). Other optimization studies involving chiral ligands, Lewis acids, co-reductants, temperature and solvents were performed.

Finally, by using a 65W compact fluorescent lamp (CFL) as the light source, the coupling of nitrone **1a** with 4-fluorobenzaldehyde was carried out in the presence of Ru(bpy)$_3$(PF$_6$)$_2$ (2.0 mol%), Sc(OTf)$_3$ (15 mol%), **L1-b** (18 mol%), and N,N,N′,N′-tetraethylethylenediamine (TEEDA, 4.0 eq) in 1,2-dichloroethane (DCE) at 0 °C for 48 h to offer the desired product **2a-1** in good yield (93% combined yield) with high diastereoselectivity and enantioselectivity (1/12 dr and 92% ee).

**Enantioselective reductive cross-coupling of nitrones with aldehydes.** With these optimized conditions in hand, we next examined the scope of nitrones. As illustrated in Table 2, symmetrical ketonitrones **1b** to **1d** gave the desired vicinal hydroxyamino alcohols with single chiral centers, respectively, in moderate to good yields with excellent enantioselectivity (Table 2, entries 2–6). The absolute configuration of **2b-3** was determined to be (S) by derivatization[39] and single-crystal X-ray diffraction analysis (see Supplementary Fig. 2). Notably, as for these symmetrical ketonitrones, dichloromethane (DCM) was better than DCE. Next, we turned our attention to the scope of aldonitrones. Compared with nitrone **1a**, more or less steric hindrance of substituents $R^3$ ($R^2$ = H) of nitrones diminished the reaction yields and stereoselectivity (Table 2, entry 1 vs. entries 7–10). The influences of N-alkyl substituent groups of nitrones were also investigated and N-benzyl nitrone **1a** provided a better result than N-methyl nitrone **1i** and N-isopropyl nitrone **1j** (Table 2, entry 1 vs. entries 11 and 12).

We also explored the scope of aldehydes and found aromatic aldehydes to be excellent partners with nitrone **1a**; aliphatic aldehydes, which were compatible with previous SmI$_2$-mediated conditions[12], are unavailable for this reaction. As for para-substituted aromatic aldehydes, the more electron-deficient one

**Table 2 Enantioselective reductive cross-coupling of nitrones with aldehydes[a]**

entry 1. **2a-1**
93% yield[b], 1/12 (anti/syn) dr[c]
92% ee (syn)[d]

entry 2[e]. **2b-1**
72% yield
97% ee

entry 3[e]. **2c**
50% yield
94% ee

entry 4[e]. **2d**
65% yield
92% ee

entry 5[e]. **2b-2**
82% yield
99% ee

entry 6[e]. **2b-3**
68% yield
94% ee

entry 7. **2e**
69% yield, 1.1/1 dr
82% ee (anti), 83% ee (syn)

entry 8. **2f-1**
72% yield, 4.1/1 dr
92% ee (anti), 42% ee (syn)

entry 9. **2g**
87% yield, 1/3.0 dr
89% ee (anti), 74% ee (syn)

entry 10. **2h**
26% yield, 1/5.0 dr
38% ee (syn)

entry 11. **2i**
41% yield, 1/5.5 dr
81% ee (syn)

entry 12. **2j**
84% yield, 1/8.0 dr
79% ee (syn)

entry 13. **2a-2**
92% yield, 1/7.2 dr
79% ee (syn)

entry 14. **2a-3**
43% yield, 1/12 dr
66% ee (syn)

entry 15. **2a-4**
87% yield, 1/6.5 dr
76% ee (syn)

entry 16. **2a-5**
99% yield, 1/5.7 dr
82% ee (syn)

entry 17. **2a-6**
62% yield, 1/15 dr
97% ee (syn)

entry 18. **2a-7**
86% yield, 1/9.8 dr
92% ee (syn)

entry 19. **2a-8**
78% yield, 1/7.7 dr
91% ee (syn)

entry 20. **2a-9**
80% yield, 1/8.5 dr
90% ee (syn)

entry 21. **2a-10**
99% yield, 1/5.3 dr
82% ee (syn)

entry 22. **2a-11**
73% yield, 1/6.2 dr
73% ee (syn)

entry 23-1[f]. **2k-1**
72% yield, 1/1.9 dr
41% ee (anti), 94% ee (syn)
entry 23-2[g]. **2k-1**
84% yield, dr < 1/20
92% ee (syn)

entry 24-1[f]. **2k-2**
73% yield, 1/1.9 dr
36% ee (anti), 92% ee (syn)
entry 24-2[g]. **2k-2**
76% yield, dr < 1/20
87% ee (syn)

entry 25-1[f]. **2k-3**
94% yield, 1/2.0 dr
49% ee (anti), 95% ee (syn)
entry 25-2[g]. **2k-3**
94% yield, dr < 1/20
89% ee (syn)

entry 26-1[f]. **2l**
72% yield, 1/2.1 dr
4% ee (anti), 97% ee (syn)
entry 26-2[g]. **2l**
64% yield, 1/12 dr
72% ee (syn)

entry 27-1[f]. **2m**
58% yield, 1/9.9 dr
97% ee (syn)
entry 27-2[g]. **2m**
47% yield, 1/20 dr
57% ee (syn)

**X-ray of 6b-3•HCl•H₂O**

**X-ray of 6a-9•HCl**

**X-ray of 6k-3•HCl**

a **General method**: Ru(bpy)₃(PF₆)₂ (2.0 mol%), Sc(OTf)₃ (15 mol%), **L1-b** (18 mol%), DCE (c 0.05 M), 65 W CFL, 0 °C, 48 h
b Isolated yield
c dr values (anti/syn) were detected from ¹H NMR or chiral HPLC analysis of crude products
d ee values were detected from chiral HPLC analysis
e DCM was used as solvent
f **Modified method 1**: Ru(bpy)₃(PF₆)₂ (2.0 mol%), Sc(OTf)₃ (15 mol%), **L1-a** (18 mol%), DIPEA (4.0 eq), DCM (c 0.05 M), 65 W CFL, −5 °C, 48 h
g **Modified method 2**: Ru(bpy)₃(PF₆)₂ (2.0 mol%), La(OTf)₃ (15 mol%), **L2-b** (30 mol%), TEEDA (4.0 eq), CH₃CN (c 0.05 M), 65 W CFL, −10 °C, 72 h

**Fig. 2** Radical clock reactions. **a** The normal cross-coupling was observed without ring opening product from ketone **3**. **b** The radical clock generated from well-designed radical clock precursor **4** was rearranged and added to nitrone **1f**

gave the higher stereoselectivity (Table 2, entries 1, 15, 18 to 21), whereas ortho- and meta-substituted aromatic aldehydes were opposite to this (Table 2, entry 14 vs. entry 17, entry 13 vs. entry 16). Moreover, a series of functional groups including thioether, secondary amide and thiophene were well tolerated in this reaction (Table 2, entries 20–22).

Furthermore, by using **L1-a** as chiral ligand and N,N-diisopropylethylamine (DIPEA) instead of TEEDA (see Methods, modified method 1), cyclic nitrone (**2k** to **2m**) also can be cross-coupled with aromatic aldehydes in moderate to excellent yields with high enantioselectivity but low diastereoselectivity (Table 2, entries 23–27; 58–94% combined yields, 92–97% ee, 1/1.9–1/9.9 dr). It is noteworthy that, by using a complex of La(OTf)$_3$ with PyBOX ligand **L2-b** as the chiral Lewis acid (see Methods, modified method 2), reductive cross-coupling of nitrone **1k** with aromatic aldehydes can also offer the desired products in good-to-excellent yields with high stereoselectivity (Table 2, entries 23–25; 76–94% combined yield, 87–92% ee, dr <1/20). Nevertheless, this modified condition was not quite compatible with nitrones **1l** and **1m**. The absolute configuration of vicinal hydroxyamino alcohols **2a-9** and **2k-3** were both determined to be (S,S) by single-crystal X-ray diffraction analysis (see Supplementary Fig. 2). Notably, aromatic ketones can also cross-couple with nitrones smoothly under racemic photocatalytic conditions to produce the desired vicinal hydroxyamino alcohols. However, following the general method or modified methods mentioned in Table 2, only traces of desired products were observed. In addition, aromatic nitrones, such as N-benzylbenzylidene amine oxide (**1o**, see Supplementary Fig. 3), always provided complex results in this reaction.

**Mechanistic investigations**. A series of experiments were carried out to get a deep insight into this reaction (see Supplementary Discussion). Control experiments showed no product was formed in the absence of photocatalyst, Lewis acid, amine reductant or light source, thus established that the reaction is synergistically catalyzed by Lewis acid and photocatalyst through a light-driven reductive process. Moreover, a radical mechanism is consistent with the phenomenon that photoreaction was entirely inhibited when 1 equivalent TEMPO (2,2,6,6-tetramethylpiperidine-1-oxyl) was added to the reaction mixture. We postulated that this photocatalytic reaction, unlike the previous SmI$_2$-mediated reactions, is initiated by the visible light excited SET reduction of aldehydes to ketyl radicals. To verify our hypothesis, radical clock

reactions of nitrone **1f** under the racemic photocatalytic condition were carried out (Fig. 2). Considering the ring opening of α-cyclopropylbenzyl radical is not a thermodynamic feasible process[40] (e.g., with 4-fluorophenyl ketone **3**, vicinal hydroxyamino alcohol **2f-2** was obtained in 27% yield as a 1/1.6 (minor/major) mixture of inseparable diastereomers, along with 72% of recovered ketone **3**), we designed and synthesized cyclopropyl-containing ketone **4** as a radical clock precursor[41]. Cyclopropyl opening of ketone **4** followed by cross-coupling with nitrone **1f** provided δ-hydroxyamino ketone **5** in 53% yield as a 1/2.0 (minor/major) mixture of diastereomers, along with 46% of recovered ketone **4**. Thus, the radical clock reactions proved that our photocatalytic reaction of nitrones with aldehydes is initiated by the SET reduction of aldehydes.

The hypothesis was supported by the cyclic voltammetry studies and density functional theory (DFT) calculations (see Supplementary Fig. 4~6 and 8~11). The coordination of 4-fluorobenzaldehyde with Sc(OTf)$_3$ in CH$_3$CN resulted in a significant reduction in the aldehyde's half-wave potential which shifted from −1.86 V to −0.62 V [vs. saturated calomel electrode (SCE)]. With the fact that nitrone has a higher affinity with Lewis acid (see Supplementary Fig. 7, the reactants' solubility studies), a complex **A** of nitrone, aldehyde and Lewis acid was supposed to be a plausible starting point of the reaction, which leads to the proposed mechanism shown in Fig. 3. Photoexcitation and reductive quenching of Ru(bpy)$_3^{2+}$ by DIPEA affords [iPr$_2$(Et)N·]$^+$ and Ru(bpy)$_3^+$ ($E_{1/2}^{II/I}$ = −1.33 V vs. SCE in MeCN[42,43]), which is sufficient to reduce complex **A** by intermolecular SET (onset potential $E_{op}$ > −0.5 V vs. SCE) and generate the radical complex **B**. Indeed, DFT calculations confirmed that the electron affinity of **A** is much higher (~ 63.0 kcal mol$^{-1}$ in free energy) than that of nitrone **1a** (~ 23.4 kcal mol$^{-1}$) and 4-fluorobenzaldehyde (~ 45.1 kcal mol$^{-1}$) in solvent, and the as-generated cross-coupling precursor **B** has spin density localized predominantly on the aldehyde moiety. Subsequently, N-radical intermediate **C** (or **C′** of anti-configuration) is formed through an analogous 6-endo-trig radical annulation and the transition state **TS$_B$** leading to a syn-configuration is predicted to be by 1.9 kcal mol$^{-1}$ favored over the anti-configuration transition state **TS$_{B'}$**. **C** upon hydrogen abstraction from [iPr$_2$(Et)N·]$^+$ affords regioselectively intermediate **D** (via **TS$_C$**) other than **D′** (via **TS$_{C'}$**). Finally, protonation of **D** gives the desired vicinal hydroxyamino alcohol **2a-1** as a major diastereomer. Moreover, DFT calculations also showed that the formation of cross-coupling precursor **B** is overwhelmingly favored over the formation of homocoupling

**Fig. 3** Proposed mechanism of this photocatalytic enantioselective reductive cross-coupling reaction. Relative Gibbs free energies (ΔG in kcal mol⁻¹ at 298 K) for key intermediates and transition states were computed at the SMD-B3LYP/DZP-level of theory

precursors, accounting well for the reaction specificity towards cross-coupling rather than homocoupling. On the basis of this mechanism, the diastereoselectivity of vicinal hydroxyamino alcohols, such as **2a-1**, can be illustrated through comparing the energy of six-member ring transition state **TS$_B$** with that of **TS$_{B'}$**. The enantioselectivity is revealed by chiral scandium complex **I**[36] involving a *Re*-to-*Re*-facial attack of the ketyl radical to nitrone **1a**.

**Concise synthesis of (+)-ephedrine and (−)-selegiline**. With the dual aim of taking further insight into this reaction and demonstrating the utility of this enantioselective radical protocol, we undertook the synthesis of ephedrine and selegiline (Fig. 4). Following the modified method 1, asymmetric reductive coupling of nitrone **1n** with benzaldehyde gave product **2n** as the major diastereomer (85% combined yield, 11/1 dr) with 94% ee[44]. After the indium-mediated reduction[39] of **2n**, vicinal amino alcohol **6n** was obtained in 90% yield. The spectral data of **6n** matched those

**Fig. 4** Concise synthesis of (+)-ephedrine **6n** and (−)-selegiline 8. A concise two-step synthesis of (1S,2-R)-(+)-ephedrine **6n** and an efficient three-step preparation of (R)-(−)-selegiline **8** have been achieved both with 70% overall yield and 94% ee

reported of ephedrine [**6n•HCl**: $[\alpha]_D^{20}$ = +30.6 (c 2.0, H$_2$O); lit.[45] for (−)-ephedrine•HCl: $[\alpha]_D^{20}$ = −34.7 (c 5.0, H$_2$O)]. The optical rotation revealed that our synthetic vicinal amino alcohol **6n** is (1 S,2 R)-(+)-ephedrine. Although the relative stereochemistry of vicinal hydroxyamino alcohol **2n** is different with the results listed in Table 2, this diastereoselectivity is supported by the DFT calculations that revealed the involvement of six-member ring transition state (see Supplementary Fig. 10). Furthermore, dehydroxylation of vicinal hydroxyamino alcohol **2n** was carried out in an aqueous solution of HCl under a mild Pd/C-catalyzed hydrogenolysis condition to provide methamphetamine hydrochloride **7**. N-propargylation of crude **7** in acetonitrile with K$_2$CO$_3$ afforded (−)-selegiline (**8**) [[$\alpha$]$_D^{25}$ = −1.02 (c 1.0, EtOH); lit.[46] for **8**: [$\alpha$]$_D^{20}$ = −1.29 (c 6.43, EtOH, >99% ee)] in 90% yield for two steps, which is a medicine used for the treatment of Parkinson's disease, depression, and senile dementia[47].

## Discussion

In summary, we demonstrated a photocatalytic enantioselective reductive cross-coupling reaction of nitrones with aromatic aldehydes via the synergistic catalysis of Ru-photocatalyst and chiral N,N'-dioxide ligand-coordinated rare earth ion. In this protocol, chiral Lewis acid represents an indispensable template for assembling to the key intermediate and triggers the asymmetric radical process to afford enantiopure vicinal hydroxyamino alcohols in moderate to excellent yields with high stereoselectivity. Taking advantage of this catalytic mechanism, unavoidable pinacol-type homocoupling side reactions in previous SmI$_2$-mediated system were entirely inhibited. Notably, chiral N,N'-dioxide ligands were used in a radical-mediated system to account for a high level of stereoselectivity. Furthermore, this reaction is operationally simple with a wide array of readily available substrates under mild condition, allowing for the step-economy synthesis of highly valuable enantiopure vicinal amino alcohols (e.g., (1S,2R)-(+)-ephedrine) and amphetamine derivatives (e.g., (R)-(−)-selegiline) rivaling those of industrial biosynthetic procedures[48]. Based on a deep insight of this reaction, we believed that a foundation has been established for further research and application of these related reactions.

## Methods

**General**. For $^1$H and $^{13}$C NMR spectra of compounds in this manuscript see Supplementary Methods. For details of the synthetic procedures and tables including detail experimental, see Supplementary Methods.

**General procedure**. An oven-dried 25 ml Schlenk tube equipped with a magnetic stir bar was added nitrone (0.30 mmol, 1.0 eq), **L1-b** (42.4 mg, 0.054 mmol, 18 mol %), Ru(bpy)$_3$(PF$_6$)$_2$ (5.2 mg, 0.006 mmol, 2.0 mol%), and Sc(OTf)$_3$ (22.1 mg, 0.045 mmol, 15 mol%) in the glove box. When the tube was sealed and removed from the glove box, DCE (or DCM) (6.0 ml) was added, followed by the aldehyde (0.45 mmol, 1.5 eq), and TEEDA (0.26 ml, 1.2 mmol, 4.0 eq). The tube was placed approximately 10 cm away from a 65 W CFL. After being stirred at 0 °C under an argon atmosphere for 48 h, the reaction mixture was filtered through a thin pad of silica gel (100–200 mesh), washed with EtOAc, and concentrated under reduced pressure. The residue was purified by flash chromatography to afford desired vicinal hydroxyamino alcohols **2a** to **2j**.

**Modified method 1**. An oven-dried 25 ml Schlenk tube equipped with a magnetic stir bar was added nitrone (0.30 mmol, 1.0 eq), **L1-a** (37.9 mg, 0.054 mmol, 18 mol %), Ru(bpy)$_3$(PF$_6$)$_2$ (5.2 mg, 0.006 mmol, 2.0 mol%), and Sc(OTf)$_3$ (22.1 mg, 0.045 mmol, 15 mol%) in the glove box. When the tube was sealed and removed from the glove box, DCM (6.0 ml) was added, followed by the aldehyde (0.45 mmol, 1.5 eq), and DIPEA (0.21 ml, 1.2 mmol, 4.0 eq). The tube was placed approximately 10 cm away from a 65 W CFL and the reaction mixture was stirred at − 5 °C under an argon atmosphere for 48 h to afford desired products **2k** to **2m**.

**Modified method 2**. An oven-dried 25 ml Schlenk tube equipped with a magnetic stir bar was added nitrone (0.30 mmol, 1.0 eq), **L2-b** (32.4 mg, 0.09 mmol, 30 mol%), Ru(bpy)$_3$(PF$_6$)$_2$ (5.2 mg, 0.006 mmol, 2.0 mol%), and La(OTf)$_3$ (26.4 mg, 0.045 mmol, 15 mol%) in the glove box. When the tube was sealed and removed from the glove box, CH$_3$CN (6.0 ml) was added, followed by the aldehyde (0.45 mmol, 1.5 eq) and TEEDA (0.26 ml, 1.2 mmol, 4.0 eq). The tube was placed approximately 10 cm away from a 65 W CFL and the reaction mixture was stirred at − 10 °C under an argon atmosphere for 72 h to afford desired products **2k** to **2m**.

**Data availability**. The crystallographic data have been deposited at the Cambridge Crystallographic Data Centre (CCDC) as CCDC 1537335 (**6a-9•HCl**), 1537337 (**6b-3•HCl**), and 1537338 (**6k-3•HCl**), and can be obtained free of charge from www.ccdc.cam.ac.uk/structures. Any further relevant data are available from the authors upon reasonable request.

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

## Acknowledgements

We are grateful to the NSF of China (21472157, 21672175, 21332007, 21273177, and 91545105), the Fundamental Research Funds for the Central Universities (No. 20720160048), the NFFTBS (No. J1310024), and the Program for Changjiang Scholars and Innovative Research Team in University (PCSIRT) for financial support. We also thank Dr. Xiao-Yu Cao of XMU for kind and helpful discussions on this paper, and Mr. Zi-Ang Nan at iChEM for recrystallization for X-ray quality crystals, X-ray crystal-lographic analysis with refinement, and valuable discussions.

## Author contributions

C.-X.Y. discovered the reaction. C.-X.Y., H.-H.L., and J.-T.C. performed the experiments and analyzed the data. Y.-P.R. and T.-T.Z. performed the chiral HPLC analysis. X.L. and Y.Y.M. performed computational studies on the mechanism. X.Z. and C.-X.Y. designed the project and wrote the manuscript. Feng's chiral ligand was suggested by P.-Q.H. X.Z., X.L., and P.-Q.H. directed the project and polished the manuscript. All of the authors discussed the results and commented on the manuscript.

## Additional information

**Competing interests:** The authors declare no competing financial interests.

