## [Peer Review File · Nature Communications]

Reviewer #1 (Remarks to the Author):

Dr Zheng and coworkers present a nice protocol for enantioselective reductive coupling of nitrones with aromatic aldehydes using chiral ligands, Lewis acids and ruthenium photocatalyst. They take advantage of the Lewis acid complexation to lower the redox potential of the aldehyde enough to enable reduction by the photocatalyst via a single electron transfer process to generate a ketyl radical. Stereoselective addition of the ketyl radical to the nitron via a 6-member transition state provides the vicinal amino alcohols enantioselectively.

This work will certainly be of interest to the chemistry community as it provides complementary approach to current technologies in the synthesis of vicinal amino alcohols. In support of their claims, the authors have provided robust characterization of their product including x-ray crystal structures. Additionally, they present mechanistic studies in support of their proposed mechanistic explanation for the reaction. I therefore, recommend publication of this paper in nature communications with some minor revisions described below.

One of claims presented as the advantage of this paper over the existing technologies is that the reaction is operationally simple, mild and high-yielding with a broad substrate scope. I agree that the reaction is milder than samarium dioxide protocols, however, I do not agree that the substrate scope presented is broad. All the substrates presented except one are benzaldehyde derivatives. Furthermore, more than 50% of these benzaldehydes are isomers of fluoro-benzaldehyde. As for the nitron scope, only alkyl nitrones are tolerated, most of them isopropyl nitrones. When cyclic nitrones are used low yield and diastereoselectivities are observed.

On the effects of steric and electronic properties on yield and enantioinduction, the authors claim that these properties play a decisive role yet the evidence provided is not convincing. Electron withdrawing para-substituents are claimed to be responsible for high yield and ee. The yields and ee's of para-fluoro-benzaldehyde compared with para-methoxy benzaldehyde are presented as evidence. More than one electron deficient benzaldehydes needs to be provided to support this claim. Furthermore, para-fluoro group is a weak electron withdrawing group with a sigma value of 0.062 ± 0.02 . Groups with stronger electron withdrawing effects such as para bromo or para-cyano groups would provide a stronger support. The authors claim that ortho and meta substituted aldehydes have lower yield and ee, yet substrates provided does not show any clear trend with steric bulk. The authors can either remove this claim or provide more evidence to support it.

Overall, the structure of the paper is good. However, figure 1 is laid out in a very confusing manner. Most of the drawings show multiple arrows arising from the same reaction center such as Figure 1, a, IV. These kinds of reaction schemes can be confusing to the reader. I would recommend that the authors show just a few clear examples for illustration on this figure.

Although the authors did a great job presenting the characterization data, some of the products are missing dr values such as: 2c, 2d, 2b-2 and 2b-3 and no explanation is provided. Additionally, I would recommend that the authors provide the characterization data of the minor diastereomers on the products with lower dr ratios such as those with dr of 2:1 or 3:1 otherwise provide an explanation for the lack of it.

Reviewer #2 (Remarks to the Author):

The manuscript by Zheng and co-workers describes the synthesis of vicinal amino alcohols from nitrones and aldehydes/ketones in enantioselective fashion using synergistic catalysis (photocatalyst and chiral ligand-coordinated Lewis acid). A Zimmerman-Traxler like transition state is proposed which is the reason for the high stereo-selectivity.

Although enantioselective protocols to access vicinal amino alcohols via photoredox pathways have been reported earlier, the present methodology is unique with good control over the selectivity and broad substrate scope.

The concept of using $\text{Sc}(\text{OTf})_3$ in combination with photoredox catalysis in which the scandium coordinated complex has a significantly lower onset potential (compared to the substrate) to facilitate the SET reduction of aldehydes to ketyl radicals is interesting.

However, the selectivity towards syn and anti-products is not clear. What is the reason for the selectivity towards syn and anti-products? Reaction with 1n give the anti as major diastereomer whereas most others give syn configuration. However, it was reasoned that the anti-coupling pathway (via transition state $\text{TSB1}'$, Fig. S10) has a smaller barrier than the syn-coupling pathway (via TSB1) resulting the inverse selectivity.

A reaction using N-benzylethanamine oxide may be useful. Comparing this result with entry 10 can give some idea what is responsible for the anti-selectivity.

Why are aliphatic aldehydes unreactive under the reaction conditions? Is it because of the difference in potentials which is retarding the SET event or does it fail to form complex A? Did the authors use any other photocatalysts (Ir(ppy)₃ Ir(III)^{*}/Ir(IV) = -1.73V; Ir(III)/Ir(II) = -2.19V etc.,) apart from Ru(bpy)₃ to check the reactivity.

α-amino dialkyl nitrones, α-amino monoalkyl nitrones used on page 5 and 6 may not be the correct term/scientific term. Please revise and use IUPAC nomenclature.

Overall, the present manuscript is very interesting and can be accepted after minor revision.

Reviewer #3 (Remarks to the Author):

The authors report an interesting synergistic catalytic approach to chiral amino alcohols from aldehydes and nitrones, employing photocatalytic activation and chiral rare-earth metal catalysis. The process is operationally simple and enables formation of synthetically useful amino alcohols in a highly stereoselective manner. They further demonstrated synthetic utility by preparing enantiopure bioactive compounds ephedrine and selegiline. The authors have also conducted extensive studies to obtain a deep understanding of the mechanistic details of the reaction using control experiments, radical clocks, DFT and cyclic voltammetry, which has led to a coherent and logical mechanistic proposal.

The disclosed reaction is of high interest to a broad chemical community. In addition to revealing extremely interesting fundamental aspects of such synergistic catalytic systems, the chemistry is of synthetic utility for preparing enantiopure amino alcohols - a widely used class of compounds in synthesis. The synthetic results in conjunction with the detailed mechanistic studies convincingly demonstrate a powerful reaction manifold that will influence the field to a large extent going forward. I would strongly recommend publication of this excellent work by Zheng, Huang and co-workers.

Recommendation: publish as is.

Reviewer #1:

Dr Zheng and coworkers present a nice protocol for enantioselective reductive coupling of nitrones with aromatic aldehydes using chiral ligands, Lewis acids and ruthenium photocatalyst. They take advantage of the Lewis acid complexation to lower the redox potential of the aldehyde enough to enable reduction by the photocatalyst via a single electron transfer process to generate a ketyl radical. Stereoselective addition of the ketyl radical to the nitron via a 6-member transition state provides the vicinal amino alcohols enantioselectively.

This work will certainly be of interest to the chemistry community as it provides complementary approach to current technologies in the synthesis of vicinal amino alcohols. In support of their claims, the authors have provided robust characterization of their product including X-ray crystal structures. Additionally, they present mechanistic studies in support of their proposed mechanistic explanation for the reaction. I therefore, recommend publication of this paper in nature communications with some minor revisions described below.

Our response: We thank the reviewer for delivering these positive feedbacks and constructive suggestions on our work.

Comment point 1: One of claims presented as the advantage of this paper over the existing technologies is that the reaction is operationally simple, mild and high-yielding with a broad substrate scope. I agree that the reaction is milder than samarium dioxide protocols, however, I do not agree that the substrate scope presented is broad. All the substrates presented except one are benzaldehyde derivatives. Furthermore, more than 50% of these benzaldehydes are isomers of fluoro-benzaldehyde. As for the nitron scope, only alkyl nitrones are tolerated, most of them isopropyl nitrones. When cyclic nitrones are used low yield and diastereoselectivities are observed.

Our response: In this manuscript, we demonstrated an asymmetric photocatalytic protocol to prepare vicinal amino alcohols with high stereoselectivity. Compared with the previously reported photocatalytic methods, our approach exhibits relatively broad scopes for both substrates (nitrones & aldehydes) corresponding to amine and alcohol moieties of vicinal amino alcohols. Compared with the previously reported SmI₂-mediated protocols, our approach can be applied to most of aromatic aldehydes (substrates), and, more importantly, overcomes the homocoupling and reduction side reactions of nitrones and aldehydes. Actually, the term “aromatic” was used to define the scope of aldehydes in the introduction and conclusion of this manuscript. To avoid misunderstanding, “**a ketyl radical**” in the abstract was revised to “**an aromatic ketyl radical**” (page 1, line 18). To show the scope of nitrones exactly, the remark “In addition, aromatic nitrones, such as *N*-benzylbenzylidene amine oxide (**1o**, see Supplementary Fig. S3), always provided complex results in this reaction.” was added on page 8, line 9 to 10.

Comment point 2: On the effects of steric and electronic properties on yield and enantioinduction, the authors claim that these properties play a decisive role yet the evidence provided is not convincing. Electron withdrawing para-substituents are claimed to be responsible for high yield and ee. The yields and ee's of para- fluoro-benzaldehyde compared with para-methoxy benzaldehyde are presented as evidence. More than one electron deficient benzaldehydes needs to be provided to support this claim. Furthermore, para-fluoro group is a weak electron withdrawing group with a sigma value of 0.062 ± 0.02 . Groups with stronger electron withdrawing effects such as para bromo or para-cyano groups would provide a stronger support. The authors claim that ortho and meta substituted aldehydes have lower yield and ee, yet substrates provided does not show any clear trend with steric bulk. The authors can either remove this claim or provide more evidence to support it.

Our response: We checked the reaction of nitrone **1a** with para-bromo benzaldehyde, and got the

desired product **2a-8** in 78% yield with $dr = 1/7.7$ and 91% ee for major *syn*-isomer. Compared with para-fluoro and para-methoxy benzaldehydes, para-bromo benzaldehyde gave a lowest yield but a better stereoselectivity. Accordingly, such new experimental data were added as entry 19 (**2a-8**) in Table 2 to further demonstrate the scope of aromatic aldehydes, and the characterization data were added on page S-15 in the Supplementary Information file. In addition, to avoid misunderstanding, the sentence “the more electron-deficient one gave the higher yield and stereoselectivity” (page 6, line 15) was revised to “the more electron-deficient one gave the higher stereoselectivity”, and the remark “These results demonstrated that both steric and electronic factors of aromatic aldehydes play decisive roles in the reaction yield and stereoselectivity.” (page 6, line 17 to 18) was removed.

Comment point 3: Overall, the structure of the paper is good. However, figure 1 is laid out in a very confusing manner. Most of the drawings show multiple arrows arising from the same reaction center such as Figure 1, a, IV. These kinds of reaction schemes can be confusing to the reader. I would recommend that the authors show just a few clear examples for illustration on this figure.

Our response: To clarify the patterns of radical coupling strategy, Figure 1, a, IV on page 3 is revised as below:

IV. Radical coupling strategy:

Convergent Carbon-Carbon Bond Formation

revised as

IV. Radical coupling strategy:

Promoted by: a) Sml_2 ; b) photocatalysis

Convergent Carbon-Carbon Bond Formation

Comment point 4: Although the authors did a great job presenting the characterization data, some of

the products are missing dr values such as: 2c, 2d, 2b-2 and 2b-3 and no explanation is provided. Additionally, I would recommend that the authors provide the characterization data of the minor diastereomers on the products with lower dr ratios such as those with dr of 2:1 or 3:1 otherwise provide an explanation for the lack of it.

Our response: Vicinal hydroxyamino alcohols **2c**, **2d**, **2b-2** and **2b-3** prepared respectively from symmetrical ketonitrones **1b**, **1c** and **1d** with benzaldehyde or para-fluoro benzaldehyde have only one chiral center. We have added the characterization data of the minor diastereomers of **2f-1** (dr = 4.1/1, page S-11), **2g** (dr = 1/3.0, page S-12), **2k-1** (dr = 1/1.9, page S-17), **2k-2** (dr = 1/1.9, page S-18), **2k-3** (dr = 1/2.0, page S-18) and **2l** (dr = 1/2.1, page S-19) in the revised Supplementary Information.

Reviewer #2:

The manuscript by Zheng and co-workers describes the synthesis of vicinal amino alcohols from nitrones and aldehydes/ketones in enantioselective fashion using synergistic catalysis (photocatalyst and chiral ligand-coordinated Lewis acid). A Zimmerman-Traxler like transition state is proposed which is the reason for the high stereo-selectivity.

Although enantioselective protocols to access vicinal amino alcohols via photoredox pathways have been reported earlier, the present methodology is unique with good control over the selectivity and broad substrate scope.

The concept of using Sc(OTf)₃ in combination with photoredox catalysis in which the scandium coordinated complex has a significantly lower onset potential (compared to the substrate) to facilitate the SET reduction of aldehydes to ketyl radicals is interesting.

Our response: We thank the reviewer for these positive remarks on our work.

Comment point 1: However, the selectivity towards *syn* and *anti*-products is not clear. What is the reason for the selectivity towards *syn* and *anti*-products? Reaction with **1n** give the *anti* as major diastereomer whereas most others give *syn* configuration. However, it was reasoned that the *anti*-coupling pathway (via transition state TS_{B1}' , Fig. S10) has a smaller barrier than the *syn*-coupling pathway (via TS_{B1}) resulting the inverse selectivity.

Our response: Compared with TS_B and $TS_{B'}$, TS_{B1} and TS_{B1}' , we speculate that the interactions (IA) between R^1 and Ar, R^3 and Ar determine the selectivity towards *syn* and *anti*-products. If IA (R^1 to Ar) is greater than IA (R^3 to Ar), *anti*-diastereomer will be major, otherwise *syn*-diastereomer will be major (see below, revised Fig. S11).

Comment point 2: A reaction using *N*-benzylethanamine oxide may be useful. Comparing this result with entry 10 can give some idea what is responsible for the *anti*-selectivity.

Our response: As suggested by the reviewer, we checked the reaction of *N*-benzylethanamine oxide **1f** with *para*-fluoro benzaldehyde and obtained *anti*-diastereomer **2f-1** as a major product (*dr* = 1.8/1 without chiral ligand, *dr* = 4.1/1 in the presence of **L1-b**). This result justifies our above-given explanation. Accordingly, **2f-1** was added in modified Table 2 (entry 8), and the characterization data was added on page S-11 in the revised Supplementary Information. On page S-32~33 in the revised Supplementary Information, the following remarks have been added to address the origin of observed stereoselectivity:

“Compare with **2a-1**, **2e~2j** and **2n**, these result suggest *anti*- or *syn*-selectivity of this radical coupling reaction may be determined by interactions between substituent R^1 or R^3 of the aldimine oxide ($R^2 = H$) and Ar group of the

aldehyde. As shown in Figure S11, when R^3 is a bulky group, the interaction between R^3 and Ar will be greater than that between R^1 and Ar, *syn*-vicinal hydroxyamino alcohol will be obtained as a major diastereomer. While R^3 is a small group, the interaction between R^1 and Ar will be greater than that between R^3 and Ar of aldehyde, *anti*-vicinal hydroxyamino alcohol will be obtained as a major diastereomer”.

Figure S11. Understanding the *syn*- or *anti*-stereoselectivity of photocatalytic reductive cross-coupling of nitrones with aromatic aldehydes.

Comment point 3: Why are aliphatic aldehydes unreactive under the reaction conditions? Is it because of the difference in potentials which is retarding the SET event or does it fail to form complex A? Did the authors use any other photocatalysts (Ir(ppy)₃ Ir(III)^{*}/Ir(IV) = -1.73V; Ir(III)/Ir(II) = -2.19V etc.,) apart from Ru(bpy)₃ to check the reactivity.

Our response: We had performed cyclic voltammetry analyses on the reduction of aliphatic aldehydes. In fact, it was found that the reduction potentials of aliphatic aldehydes were lower than -2.20 V, even in the presence of Lewis acid and nitrones, which let us estimate that photocatalysts Ir(ppy)₃ cannot work for SET reduction of aliphatic aldehydes.

We also checked this hypothesis by several control experiments, and found photocatalyst Ir(ppy)₂(dtbbpy)(PF₆) can give vicinal hydroxyamino alcohol **2a-1** in 56% yield, but also was not working for aliphatic aldehyde (*i*-PrCHO). To provide more detailed supporting information, a

sentence “Other photocatalysts, such as Ir(ppy)₃ and Ir(ppy)₂(dtbbpy)(PF₆) also were used under this condition, but just the latter can give desired product in moderate yield (56%).” was added on page S-5, and corresponding entries 3 and 4 were added to Table S2 in the revised Supplementary Information.

Comment point 4: α -amino dialkyl nitrones, α -amino monoalkyl nitrones used on page 5 and 6 may not be the correct term/scientific term. Please revise and use IUPAC nomenclature.

Our response: In IUPAC nomenclature, a “nitron” is named as an imine oxide. Accordingly, “ α -amino dialkyl nitrones” should be termed “symmetrical ketimine-type of nitrones”, and “ α -amino monoalkyl nitrones” should be termed “aldimine-type of nitrones”. Also, “ketonitron” and “aldonitron” were often used to refer to ketimine-type of nitrones and aldimine-type of nitrones, respectively. Therefore, on page 6, the term “ α -amino dialkyl nitrones” was revised as “symmetrical ketonitrones”, and “ α -amino monoalkyl nitrones” was revised as “aldonitrones”.

Overall, the present manuscript is very interesting and can be accepted after minor revision.

Reviewer #3:

The authors report an interesting synergistic catalytic approach to chiral amino alcohols from aldehydes and nitrones, employing photocatalytic activation and chiral rare-earth metal catalysis. The process is operationally simple and enables formation of synthetically useful amino alcohols in a highly stereoselective manner. They further demonstrated synthetic utility by preparing enantiopure bioactive compounds ephedrine and selegiline. The authors have also conducted extensive studies to obtain a deep understanding of the mechanistic details of the reaction using control experiments,

radical clocks, DFT and cyclic voltammetry, which has led to a coherent and logical mechanistic proposal.

The disclosed reaction is of high interest to a broad chemical community. In addition to revealing extremely interesting fundamental aspects of such synergistic catalytic systems, the chemistry is of synthetic utility for preparing enantiopure amino alcohols - a widely used class of compounds in synthesis. The synthetic results in conjunction with the detailed mechanistic studies convincingly demonstrate a powerful reaction manifold that will influence the field to a large extent going forward. I would strongly recommend publication of this excellent work by Zheng, Huang and co-workers. Recommendation: publish as is.

Our response: We thank the reviewer for these positive remarks and recognition for our work.

Other minor revisions:

1. On page 1, line 6~8, “²Collaborative Innovation Center of Chemistry for Energy Materials & State Key Laboratory of Physical Chemistry of Solid Surfaces, Xiamen University, Xiamen, Fujian 361005, China.” was add as the second institution.
2. On page 1, line 11, the first sentence in the abstract, the term “**Enantiopure vicinal amino alcohols and their derivatives**” was changed to “**Enantiopure vicinal amino alcohols and derivatives**”.
3. All changes have been highlighted with yellow in the revised manuscript and Supplementary Information. Because **2f-1** (Table 2, entry 8) and **2a-8** (Table 2, entry 19) were added, some serial numbers of compounds have been changed respectively, i.e., “**1f, 1g, 1h, 1i, 1j, 1k, 1l**” to “**1g, 1h, 1i, 1j, 1k, 1l, 1m**”, “**2f, 2g, 2h, 2i, 2j, 2k, 2l**” to “**2g, 2h, 2i, 2j, 2k, 2l, 2m**”, “**1m**” to “**1f**”, “**2m**” to “**2f-2**”, and “**2a-8, 9, 10**” to “**2a-9, 10, 11**”.

4. On page 5, line 2: “see Supplementary Information for details” was changed to “see Supplementary Note 1 for details”.
5. On page 6, line 8: “ α -amino alkyls” was changed to “substituents R^3 ($R^2 = H$) of aldonitrones”.
6. On page 6, line 19: “thiol” was revised to “thioether”.
7. On page 9, line 18: “ $E_{op} < -0.5 V$ ” was revised to “ $E_{op} > -0.5 V$ ”.

Reviewer #1 (Remarks to the Author):

The authors have sufficiently addressed the reviewers comments. I believe the updated manuscript can be accepted by nature communications as is.